# Federated Learning for Feature Generalization with Convex Constraints

Dongwon Kim[1]   Donghee Kim[1]   Sung Kuk Shyn[2]   Kwangsu Kim[1]

## Abstract

Federated learning (FL) often struggles with generalization due to heterogeneous client data. Local models are prone to overfitting their local data distributions, and even transferable features can be distorted during aggregation. To address these challenges, we propose FedCONST, an approach that adaptively modulates update magnitudes based on the global model's parameter strength. This prevents over-emphasizing well-learned parameters while reinforcing underdeveloped ones. Specifically, FedCONST employs linear convex constraints to ensure training stability and preserve locally learned generalization capabilities during aggregation. A Gradient Signal to Noise Ratio (GSNR) analysis further validates FedCONST's effectiveness in enhancing feature transferability and robustness. As a result, FedCONST effectively aligns local and global objectives, mitigating overfitting and promoting stronger generalization across diverse FL environments, achieving state-of-the-art performance.

## 1. Introduction

Federated Learning (FL)(McMahan et al., 2017) has emerged as a promising paradigm that enables multiple clients to collaboratively learn a shared model while keeping their data localized. A pivotal challenge in FL arises from the sparse and heterogeneous data distribution across clients, which leads to significant problems on the performance of a global model. (1) Local models often overfit their own distributions, limiting their capacity to generalize across the entire data distribution(Qu et al., 2022; Mendieta

[1]Department of Computer Science and Engineering, University of Sungkyunkwan, Suwon, Korea [2]Kim Jaechul Graduate School of AI, Korea Advanced Institute of Science and Technology (KAIST), Daejeon, Korea. Correspondence to: Dongwon Kim <kdwaha@skku.edu>, Kwangsu Kim <kim.kwangsu@skku.edu>.

*Proceedings of the 42nd International Conference on Machine Learning*, Vancouver, Canada. PMLR 267, 2025. Copyright 2025 by the author(s).

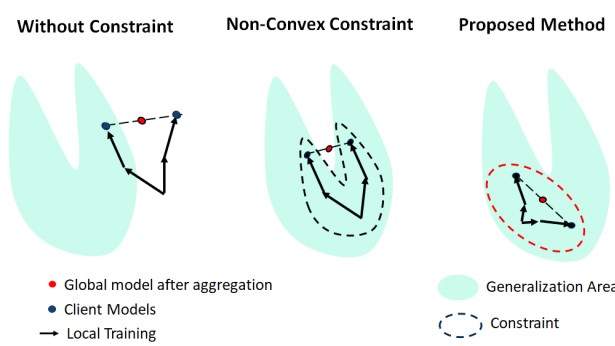

*Figure 1.* Illustration of the parameter space in FL. (1) Vanilla FL drives the optimization process away from the generalization area. (2) Optimization with non-convex constraints stabilizes the training process within the generalization area but may cause a loss of generalization during aggregation. (3) Our convex constraints stabilize the training process and align the aggregation with the generalization area, ensuring improved global generalization.

et al., 2022). (2) Even when some clients learn features that could generalize, these can become distorted during aggregation, misaligning them with global objectives and undermining performance(Lee & Yoon, 2024).

Previous approaches, including regularization(An et al., 2024; Li et al., 2021a; 2020), normalization (Li et al., 2021b; Andreux et al., 2020; Wang et al., 2023) and correction (Acar et al., 2021; Karimireddy et al., 2020; Varno et al., 2022) aim to mitigate these issues by aligning local model updates with the global model. However, by focusing solely on preserving global model information, these methods have neglected to ensure generalization, inevitably resulting in overfitting under limited data conditions.

Recently, some studies have turned their attention to generalization of local learning. FedAlign(Mendieta et al., 2022) and FedSAM(Qu et al., 2022) focused on the generalization of local learning employing generalization term on loss. However, by failing to preserve coherent optimization objectives across clients during aggregation, these methods allow the model's generalization capabilities to become distorted, ultimately degrading its overall performance.

In light of these issues, we ask: *What kind of constraint*

*enhances feature generalization during local training while remaining unaffected by the aggregation process?*

To answer this question, We propose FedCONST (Federated Learning with Convex Constraints for Global Model Generalization). FedCONST applies client-consistent convex constraints derived from the global model's weight magnitudes, which serve as proxies for feature importance across the entire data distribution. Concretely, well-learned (stable) features in the global model are constrained to remain close during local updates, while under-learned (unstable) features are emphasized for further training. Because the constraints are convex and shared among all clients, they preserve the intended generalization effect after aggregation as desribed in Figure 1.

Our method is inspired by insights from a Domain Generalization (DG) method(Michalkiewicz et al., 2023), where strong features (often measured by gradient statistics such as Gradient Signal-to-Noise Ratio, GSNR) are crucial to robust performance across diverse domains. Directly tracking GSNR is typically infeasible in FL due to communication and computational bottlenecks. Instead, we show that global weight magnitudes are reliable proxy for feature strength, aligning well with the GSNR perspective. This design choice makes FedCONST simple, communication-efficient, and broadly applicable.

In this paper, we provide the motivation behind our work and a theoretical foundation for our method. We demonstrate higher stability in local training through reduced gradient variance and improved convexity of the global model loss landscape. Consequently, our experiments show that FedCONST significantly outperforms existing FL methods in various models, datasets, and levels of heterogeneity in both cross-device and cross-silo settings, while maintaining high computational and communication efficiency.

In summary, our contributions are as follows.

- We propose a simple, yet effective approach that retains well-learned features while focusing on under-learned ones. This framework introduces new insights into how generalization can be enhanced in FL.

- Our theoretical and empirical analyses guarantee that our method boosts generalization by imposing more updates with larger probabilities to under-learned features.

- We validate the effectiveness of FedCONST with a wide range of dataset and model architectures and show that it significantly outperforms existing FL methods with SOTA performance.

## 2. Related work

### 2.1. Federated Learning on Non-IID Data

The challenge of non-IID data across clients leads to unstable local learning diverge the global model from consistent optima. To mitigate these issues, regularization methods have been prominently adopted. Techniques like FedProx (Li et al., 2020), FedMRUR(An et al., 2024), and MOON(Li et al., 2021a) incorporate explicit regularization mechanisms to align local updates with global objectives more effectively. These approaches ensure that the local models have common features for global objectives. However, common features are often spurious as well, making global model suffer from the overfitting problem.

To directly address the challenges posed by heterogeneous gradients, correction methods like FedDyn(Acar et al., 2021), AdaBest(Varno et al., 2022), and SCAFFOLD(Karimireddy et al., 2020) introduce correction terms that aim to align client updates with the global model. These strategies utilize stateful operations to align client updates on FL environment with limited local data. Aside from the risk on alignment of overfitting problem, this stateful operations require extra communications.

On the other hand, methods such as FedSAM(Qu et al., 2022) and FedAlign(Mendieta et al., 2022) focus on enhancing generalization across clients without imposing restrictions for alignment with global objectives. These techniques prioritize a generalization of local training on limited data but unaligned approach to handle the heterogeneity inherent in FL.

### 2.2. Generalization of Neural Networks

To analyze generalization performance during the training process, a study proposed the concept of One Step Generalization Ratio (OSGR)(Liu et al.). OSGR is defined as the ratio between the decrease in loss on test data and on training data:

$$R_{Z,n} = \frac{\mathbb{E}_{D,D' \sim Z^n}[\Delta\mathcal{L}_{D'}]}{\mathbb{E}_{D \sim Z^n}[\Delta\mathcal{L}_D]}, \tag{1}$$

where $\Delta\mathcal{L}_{D'}$ and $\Delta\mathcal{L}_D$ denote the decrease in loss on training data $D$ and test data $D'$ within a single optimization step.

To facilitate the prediction of OSGR during training, the authors further propose the following theorem.

**Proposition 2.1** (From Paper (Liu et al.))**.** *The generalization of gradient updates can be expressed using the following OSGR value:*

$$R_{Z,n} = 1 - \frac{1}{n}\sum_j W_j \frac{1}{\frac{g_j^2}{\rho_j^2} + \frac{1}{n}}, \tag{2}$$

*where $n$ is the number of samples, $g_j^2$ is the squared gradient magnitude for feature $j$ and $\rho_j^2$ is the corresponding noise variance, and The weight $W_j$ (satisfying $\sum_j W_j = 1$) is a weighting term.*

Proposition 2.1 indicates that features with higher Gradient Signal-to-Noise Ratios(GSNR), defined by:

$$r_j = \frac{g_j^2}{\rho_j^2} \tag{3}$$

yield larger values of OSGR, thereby contributing more significantly to generalization performance.

Based on these insights, a Paper (Michalkiewicz et al., 2023) proposed a GSNR-based dropout method for DG tasks, aiming to enhance robustness by preserving parameters with higher GSNR values while promoting updates in those with lower GSNR. However, the distributed nature of FL complicates gradient collection and aggregation across clients.

## 3. Our Approach

To begin with, we define standard learning process is to train a deep neural network $f(x; W)$, where $f : X \to Y$ is a neural network with $L$ neural layers :

$$W = \{w^1, w^2, \dots, w^L\} \tag{4}$$

Especially, We are interested in training each weight $w^l = \{w_{c,1}^l, \dots, w_{c,s}^l\}$ with respect to the corresponding feature/channel $i$ on input layer $l$.

### 3.1. Federated Learning

We consider the standard FL that trains a model collaboratively from decentralized client devices. For every client, denoted as $m$ in the set $M$, there are $N_m$ training samples given by pairs $(x_i,\ y_i)$ for $i = 1$ to $N_m$. Here, $x_i$ represents the image from a set $X$ and $y_i$ is the corresponding label from set $Y$. These pairs are independently and identically distributed, drawn from a distribution specific to the device, symbolized as $D_m(x, y)$ when $D = \bigcup_m D_m$. With this setting, we follow the framework of FedAvg follows:

$$\mathcal{L}(W) = \sum_{m \in M} \frac{|D_m|}{|D|} \mathcal{L}_m(W)$$
$$\text{where} \quad \mathcal{L}_m(W) = \mathbb{E}_{(x_i, y_i) \sim D_m} [\mathcal{L}(x_i, y_i; W)] \tag{5}$$

The global objective, denoted as $\mathcal{L}$, can be broken down into individual empirical loss $\mathcal{L}_m$ specific to each client data. Because of the separation of clients' data, $\mathcal{L}$ can not be optimized directly. FedAvg addresses this challenge by alternating between local training on each client's dataset and a global aggregation step.

**Client Training** Pivoting to the FedAvg blueprint, the local training trajectory is captured as:

$$w_m^k = w^k - \eta \sum_{t \in T} g_{m,\ t}^k = w^k + \Delta w_m^k \tag{6}$$

Here, $w^k$ typifies the global model during global round $k$, $g_{m,\ t}^k$ is the gradient corresponding to client m at timestep t of the $k^{th}$ global round, and $\eta$ denotes the learning rate.

**Aggregation** Transitioning to the global aggregation phase, the mechanics unfold as:

$$W^{k+1} = \frac{1}{M} \sum_{m \in M} w_m^k = W^k + \frac{1}{M} \sum_{m \in M} \Delta w_m^k$$
$$= W^k + \Delta w^k \tag{7}$$

In this matrix, $\Delta w^k$ epitomizes the average model updates accumulated from all clients during the $k_{th}$ global round.

### 3.2. FedCONST:Federated Learning for Feature Generalization with Convex Constraints

For promoting generalization in FL, we begin by proposing a conjecture and two principles to design our constraints. Next, we demonstrate that our method adheres to these principles. Finally, we present evidence to support the validity of the conjecture.

#### 3.2.1. CONSTRAINED OPTIMIZATION FOR CLIENT TRAINING

We propose a Constrained Weight Optimization framework that transfers the key insight of Feature Strength—initially introduced in DG methods—to a FL scenario. Our approach is motivated by Conjecture 1, which states that the magnitude of a weight can serve as a proxy for how well a feature is learned.

**Conjecture 1** *Large weights indicate well-learned (strong) features. Small weights signify weaker features requiring additional training.*

By leveraging this conjecture, we can selectively preserve strong features while reinforcing weaker ones, preserving the core principle of DG methods without direct access to each client's detailed gradients. To operationalize Conjecture 1 within federated learning, we introduce a constrained optimization framework:

$$\min_W \mathcal{L}_m(W)$$
$$\text{s.t. } {G_c^l}^\top (w_c^l - G_c^l) = 0, \quad \mathbf{1}^\top w_c^l = 0, \quad \forall c, l \tag{8}$$

Here, $w_i^l$ represents the weight vector for the $i$-th output feature in the $l$-th layer of a client model, and $G_c^l$ means

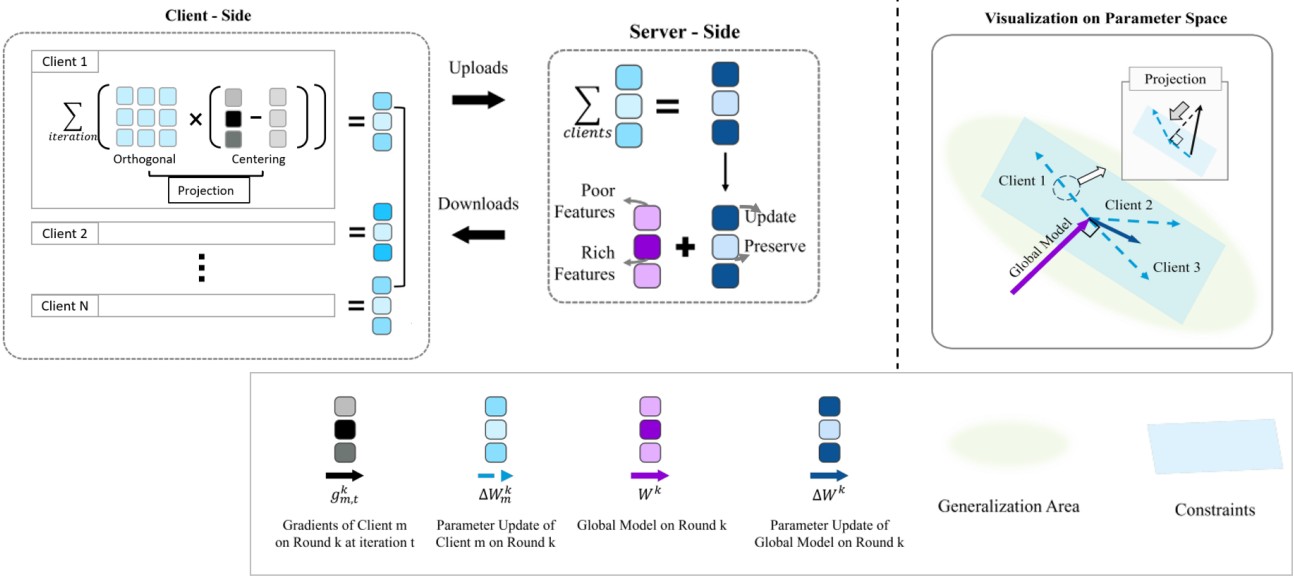

*Figure 2.* Schematic representation of FedCONST: Local learning on clients with weight constrained optimization to preserve robust paramter of the global model. Aggregation phase where convex constraints guide diverse client models toward a common, optimal representation, facilitating better alignment and performance of the global model.

the corresponding weight vector of global model. $1$ is an all-one vector.

This design is originated form the insight that constraints should adjust feature learning while maintaining generalization ability after the aggregation phase. This can be summarized as the following condition:

**Condition 1. (Feature Adjustment)** *The constraints boost weak features and preserve already strong features for generalization.*

**Condition 2. (Convex Constraints)** *The constraints should be conserved after aggregation to preserve generalization ability.*

Our design consists of two constraints, the centralization constraint and the orthogonal constraint. The centralization constraint($1^\top w_c^l = 0$ on Equation (8)) ensures that the total update impact on a feature is equalized, while the orthogonal constraint($G_c^{l\top}(w_c^l - G_c^l) = 0$ on Equation (8)) mitigates redundant updates to already strong signals. Together, these constraints satisfy Condition 1 by promoting generalization during local training and Condition 2 by being linear and convex.

### 3.2.2. FEATURE ADJUSTMENT

In this section, we justify our constraints prevent redundant training on strong features. We assume that weight changes due to gradient updates on parameters $W_c^l = \{W_{c,1}^l, \ldots, W_{c,s}^l\}$ with respect to the corresponding fea-

ture/channel $i$ follow a spherical Gaussian distribution:

$$\Delta W_{c,q} \sim \mathcal{N}\left(0, \sigma^2 I\right), \text{where } q = 1, 2, \ldots s \qquad (9)$$

$$\sigma^2 = \frac{\sum_q \left(\Delta W_{c,q}^l\right)^2}{n - 1}. \qquad (10)$$

We project the weight changes onto the hyperplane orthogonal to the initial client weight vector $W_c^l$, which is initialized to match the global parameter of $G_c^l$, and define the projection as follows:

$$P = I - uu^\top, \quad u = \frac{W_c^l}{\|W_c^l\|_2}, \qquad (11)$$

the projected weight update is:

$$\Delta w_\perp = P\Delta w,$$
$$\text{Cov}(\Delta w_\perp) = P \cdot \text{Cov}(\Delta w) \cdot P^\top = \sigma^2 P \qquad (12)$$

The resulting variance aligns with:

$$\text{Var}(\Delta W_{c,q\perp}^l) = \sigma^2 \left(1 - \frac{W_{c,q}^l{}^2}{\|W_c^l\|_2^2}\right). \qquad (13)$$

So these discussion can be summarized as the following proposition:

**Theorem 3.1** (Feature-Preserving Updates under Centering and Orthogonality)**.** *If we impose centering and orthogonality constraints, and if $W_{c,i}^l \leq W_{c,j}^l$, then*

$$Var(\Delta W_{c,i}^l) \geq Var(\Delta W_{c,j}^l),$$

*which means that updates are inversely correlated with the weight size, i.e.,*

$$\Pr(|\Delta W_{c,i}^l| \geq |\Delta W_{c,j}^l|) \geq \Pr(|\Delta W_{c,i}^l| \leq |\Delta W_{c,j}^l|).$$

Therefore, our constraints ensure that updates have an inverse correlation with weight size, promoting stability and avoiding overfitting to features with larger weights.

### 3.2.3. CONVEX CONSTRAINTS

Each client optimizes its local model independently on private data, and these models are then aggregated to form a global model without alignment. To address this misalignment, we refine our constraints to be convex, ensuring they satisfy the conditions for generalization both during local training and after aggregation. Although these constraints are applied per weight vector for channel/feature $c$ of layer $l$, we omit $c$ and $l$ in the notation for brevity and to focus on the convex characteristic.

**Centralization Constraint(**$1^\top w_c^l = 0$ **on Equation (8))**
The Centralization Constraint is specifically applied to the gradient of each client's local model to stabilize the local training process by maintaining the mean to be zero. If each gradient of every client is centralized as:

$$1^\top g_{m,t}^k = 0 \tag{14}$$

then it results in

$$-\eta \sum_{t \in T} 1^\top g_{m,t}^k = 1^\top (-\eta \sum_{t \in T} g_{m,t}^k) = 1^\top \Delta w_m^k = 0 \tag{15}$$

Consequently, the total change of a client model during local training becomes zero, ensuring unbiased weight on client models.

By Equation (15), the global update adhering to the Centralization Constraint is represented as:

$$\frac{1}{M} \sum_{m \in M} 1^\top \Delta w_m^k = 1^\top \Delta w^k = 0 \tag{16}$$

In Algorithm 1, to apply the centralization constraint $1^\top w_i^l = 0$ in practice, we define the following centering function:

$$C(w) = w - \frac{1}{n} 1^\top w \tag{17}$$

where $n$ denotes the number of parameters in $w$. This function is directly used in our algorithm to enforce the centralization constraint during local updates.

**Orthogonal Constraint(**$G_c^{l\top}(w_c^l - G_c^l) = 0$ **on Equation (8))**
The orthogonal constraint align the parameters of local and global models onto the same hyperplane orthogonal to the initial global model at each round, mitigating the grow in

the strong features. This adjustment translates the constraint such that the local gradient is orthogonal to the initial global model:

$$(w^k)^\top g_{m,t}^k = 0 \tag{18}$$

then it results in

$$-\eta \sum_{t \in T} (w^k)^\top g_{m,t}^k = (w^k)^\top (-\eta \sum_{t \in T} g_{m,t}^k)$$
$$= (w^k)^\top \Delta w_m^k = 0 \tag{19}$$

During aggregation, Equation (19) aligns the global update as:

$$\frac{1}{M} \sum_{m \in M} (w^k)^\top \Delta w_m^k = (w^k)^\top \Delta w^k = 0 \tag{20}$$

In Algorithm 1, we apply the projection operator $P_{w^k}$ to each update direction to ensure that local updates remain orthogonal to the initial global parameter $p = w^k / \|w^k\|$.

$$P_{w^k}(w) = (I - pp^\top)w, \tag{21}$$

where $I$ is the identity matrix, and $pp^\top$ is the outer product of $p$ with itself. This projection operator is used in our algorithm to enforce the orthogonality constraint by projecting updates onto the tangent space of the global model direction.

### 3.3. Weight Size as a Feature Strength

In this section, we discuss about validity of conjecture 1. Fortunately, the paper (Liu et al.) also provided a detailed analysis of the relationship between GSNR and weight size, indicating a positive correlation. They considered a fully connected network with parameters

$$\theta = \{W^1, \ldots, W^{l_{\max}}\} \tag{22}$$

where $W^l, b^l$ are the weight matrix and bias of the first layer, respectively, and so on. The activations of the $l$-th layer are denoted by

$$a^l = \{a_s^l(\theta^{l-1})\} \tag{23}$$

where $s$ is the index for nodes/channels, and

$$\theta^{l-1} = \{W^1, \ldots, W^{l-1}\} \tag{24}$$

is the collection of parameters in the layers before $l$. The forward pass on data sample $i$, where $\{a_s^l(\theta^{l-1})\}$ is multiplied by the weight matrix $W^l$, is defined as:

$$o_{i,c}^l = \sum_s W_{c,s}^l a_{i,s}^l(\theta^{l-1}) \tag{25}$$

where $o^l = \{o_{i,c}^l\}$ is the output for the $i$-th data sample on the $l$-th layer, and $c$ is the index for nodes/channels. We use

$g^l$ to denote the average gradient of weights of the $l$-th layer $W^l$, i.e.,

$$g^l = \frac{1}{n} \sum_{i=1}^{n} \frac{\partial \mathcal{L}_i}{\partial W^{(l)}} \qquad (26)$$

where $\mathcal{L}_i$ is the loss of the $i$-th sample.

In this setting, the authors showed the following correlation between gradient change and gradient norm size:

$$\Delta g_{s,c}^l = -\frac{\lambda}{n^2} \sum_{\theta_j \in \theta^{l-1}} W_{s,c}^l \left( \frac{1}{n} \sum_{i=1}^{n} \frac{\partial \mathcal{L}_i}{\partial o_{i,c}^l} \frac{\partial o_{i,c}^l}{\partial \theta_j} \right)^2 \qquad (27)$$
$$+ \text{ other terms}$$

where $\lambda$ is the learning rate, assumed to be small enough.

This expression implies that if the gradient change $\Delta g_{s,c}^l$ and the corresponding weight $W_{s,c}^l$ have different signs, they contribute to a more stable state by constructing positive feedback during training, which increases the size of both values. Conversely, if they have the same signs, a negative feedback loop during training decreases the size of both values until one of them changes its sign, resulting in a stable state.

Therefore, considering only stable states, the size of the weight $W_{s,c}^l$ is directly correlated with the gradient change $\Delta g_{s,c}^l$, positively affecting the GSNR value of $r_j = \frac{g_j^2}{\rho_j^2}$ on Equation (3). Here, we decided to use the size of $W_{s,c}^l$ of the global model instead of collecting $g_{s,c}^l$ statistics to calculate GSNR values for the entire dataset. As a result, rather than utilizing GSNR as an indicator of feature strength, which requires collecting gradients from each client, we adopt the weight magnitude of the global model as a proxy for feature strength.

### 3.4. Training Process

As shown in Algorithm 1, FedCONST merely changes the local learning process to constrained optimization on convex linear constraints for Global Model Generalization. This approach has two main advantages: (1) Using convex constraints based on common global model, we align local training across client models without additional communication cost in stateless manner. (2) By employing the weight size to estimate the generalizability of the features, we ensure the generalizability of the FL process. Moreover, this insight is well motivated by GSNR based analysis.

## 4. Experiments

### 4.1. Experimental Setup

We conducted experiments using the CIFAR-10 and CIFAR-100 datasets (Krizhevsky et al., 2009). Our experiments

---

**Algorithm 1** Training procedure of FedCONST

**Input:** Batch size $B$, communication rounds $K$, number of clients $M$, local steps $T$, dataset $D = \bigcup_{m \in [M]} D_m$
**Output:** Global model parameters $w^K$
**Server executes:**
  Initialize $w^0$ with He Initialization
  **for** $k = 0, \ldots, K-1$ **do**
    **for** $m = 1, \ldots, M$ **in parallel do**
      Send $w^k$ to client $m$
      $w_m^{k+1} \leftarrow$ **FedCONST: Client executes**$(m, w^k)$
    **end**
    $w^{k+1} \leftarrow \sum_{m \in [M]} \frac{|D_m|}{|D|} w_m^{k+1}$
  **end**
**return** $w^K$
  **FedCONST: Client executes**$(m, w^k)$:
  Assign global model to the local model $w_m^k \leftarrow w^k$
  **for** each local epoch $t = 1, \ldots, T$ **do**
    **for** batch $(x_{m,1:B}, y_{m,1:B}) \in D_m$ **do**
      Per layer $l$ and channel/feature $c$,
        Center gradient: $g_{m,t}^k \leftarrow \mathbf{C}(g_{m,t}^k)$
        Project gradient: $g_{m,t}^k \leftarrow P_{w^k}(g_{m,t}^k)$
        Apply update: $w_m^k \leftarrow w_m^k - \eta g_{m,t}^k$
    **end**
  **end**
**return** $w_m^{k+1}$ to server

---

spanned both cross-silo and cross-device settings. For the cross-silo setup, we involved a total of 10 clients, while in the cross-device setting, 10% of clients were randomly selected from a pool of 50 or 100 participants. The data distribution among clients was governed by a Dirichlet distribution, with the $\alpha$ value determining the degree of heterogeneity; a lower $\alpha$ value corresponds to a more heterogeneous distribution. For an extremely heterogeneous environment, we used a Dirichlet $\alpha$ of 0.2 with 10 local training epochs, while a more typical environment utilized an alpha of 0.5 with 5 local training epochs. Our tests were conducted on both the LeNet-5 and ResNet-18 architectures. More detailed settings of experiments are in the supplementary materials.

### 4.2. Performance Comparison

As detailed in Table 1, the LeNet-5 model consistently outperformed the existing algorithms across all settings. As shown in Table 1, our approach surpassed the performance of existing algorithms in all settings on CIFAR-10 and demonstrated commendable performance on CIFAR-100. Table 1 further reveals that in a cross-device setting, our algorithm consistently exceeded the performance of the other algorithms. Moreover, performance enhancement was

*Table 1.* Top-1 test accuracy (%) comparison of LeNet-5 and ResNet-18 models under Cross-Device and Cross-Silo settings. The numbers inside the parentheses represent the accuracy differences when Constraint was applied to the training of client models.

| | | CROSS-DEVICE | CROSS-SILO | | |
| | | CIFAR-10 | CIFAR-10 | CIFAR-10 | CIFAR-100 |
| MODEL | ALGORITHM | $\alpha = 0.5$ | $\alpha = 0.2$ | $\alpha = 0.5$ | $\alpha = 0.5$ |
| | FEDAVG | 46.12 | 46.42 | 53.12 | 17.46 |
| | FEDAVG + CONST | **54.28** (+8.16) | 54.79 (+8.37) | 59.66 (+6.54) | **26.86** (+9.40) |
| | FEDPROX | 45.58 | 45.27 | 55.15 | 18.42 |
| | FEDPROX + CONST | 53.09 (+7.51) | 56.18 (+10.91) | 60.70 (+5.55) | 26.78 (+8.36) |
| LENET-5 | MOON | 43.89 | 46.66 | 55.79 | 18.72 |
| | MOON + CONST | 48.66 (+4.77) | 52.88 (+6.22) | 59.86 (+4.07) | 26.76 (+8.04) |
| | SCAFFOLD | 45.66 | 45.67 | 52.74 | 17.66 |
| | SCAFFOLD + CONST | 53.82 (+8.16) | **56.62** (+10.95) | **63.03** (+10.29) | 26.74 (+9.08) |
| | FEDDYN | 44.93 | 48.05 | 51.05 | 16.79 |
| | FEDDYN + CONST | 54.07 (+9.14) | 55.67 (+7.62) | 59.76 (+8.71) | 27.14 (+10.35) |
| | FEDAVG | 54.07 | 57.04 | 64.25 | 33.51 |
| | FEDAVG + CONST | 66.51 (+12.44) | 68.41 (+11.37) | 72.44 (+8.19) | 36.82 (+3.31) |
| | FEDPROX | 56.79 | 53.92 | 64.51 | 34.11 |
| | FEDPROX + CONST | 63.51 (+6.72) | 68.07 (+14.15) | 71.96 (+7.45) | 36.56 (+2.45) |
| | MOON | 57.84 | 51.51 | 68.45 | 35.19 |
| RESNET-18 | MOON + CONST | **66.94** (+9.10) | 62.52 (+11.01) | 71.84 (+3.39) | 36.80 (+1.61) |
| | SCAFFOLD | 56.47 | 59.30 | 64.50 | 37.18 |
| | SCAFFOLD + CONST | 63.49 (+7.02) | 68.63 (+9.33) | **75.09** (+10.59) | 38.93 (+1.75) |
| | FEDDYN | 52.64 | 55.09 | 65.50 | 35.07 |
| | FEDDYN + CONST | 64.29 (+11.65) | 66.00 (+10.91) | 71.76 (+6.26) | 37.22 (+2.15) |
| | FEDSAM | 62.52 | 61.35 | 69.45 | 38.43 |
| | FEDSAM + CONST | 63.45 (+0.93) | **68.87** (+7.52) | 72.64 (+3.19) | **39.61** (+1.18) |

observed when constraints were applied to algorithms representative of either alignment or local learning generalization methods. This indicates that our method fills the gaps and enhances areas where existing methods fall short, leading to a more robust and efficient FL process.

### 4.3. Experiment Analysis

#### 4.3.1. STABILITY ON LOCAL TRAINING

Figure 3(a) demonstrates that our method reduces gradient variance compared to FedAvg, thereby stabilizing local updates in environments with sparse data. Moreover, Figure 3(b) illustrates that our method enhances drift diversity, defined as $\frac{\sum_{m \in M} \frac{|D_m|}{|D|} \|\Delta w_m^k\|_2^2}{\|\Delta w^{k+1}\|_2^2}$ (Li et al., 2023) ensuring that each client learns effectively and can fully reflect its own unique data characteristics even with constraints. The increase in drift diversity compared to FedAvg indicates that our method results in larger client updates after local training, enabling effective model updates despite sparse data or parameter constraints. Overall, these results demonstrate that our approach effectively mitigates instability in local training, ensuring sufficient updates within the imposed constraints.

#### 4.3.2. MODEL ALIGNMENT ACROSS CLIENT

In our analysis of loss landscape, we shifted our analytical focus from sharpness-based metrics to convexity-based metrics. Previous research primarily utilized the maximum eigenvalue $\lambda_{max}$ of the model's Hessian matrix to measure sharpness, which correlates with generalization performance (Mendieta et al., 2022). However, our analysis suggests that observing the ratio of the absolute maximum eigenvalue to the minimum eigenvalue $|\lambda_{max}/\lambda_{min}|$ — a measure of convexity(Rangwani et al., 2022) — more effectively captures the essence of model alignment.

*Table 2.* **Loss Landscape Convexness.** The metric $C_{convex}$ represents the $|\lambda_{max}/\lambda_{min}|$ values of ResNet-18 model in Cross-Device settings. The term $H_{trace}$ denotes Hessian trace value.

| | W/O CONSTRAINTS | | CONSTRAINTS | |
| ALGORITHM | $C_{convex}$ | $H_{trace}$ | $C_{convex}$ | $H_{trace}$ |
| FEDAVG | 2.466 | -4951 | 31.7 | 10102 |
| FEDPROX | 2.739 | -3873 | 23.09 | 12294 |
| MOON | 3.015 | -3416 | 16.09 | 9640 |
| SCAFFOLD | 2.914 | -3245 | 21.81 | 9145 |
| FEDDYN | 2.16 | -4590 | 12.82 | 9121 |
| FEDCONST | **31.7** | **10102** | - | - |

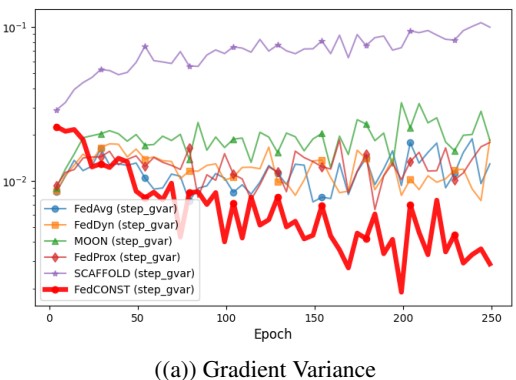

((a)) Gradient Variance

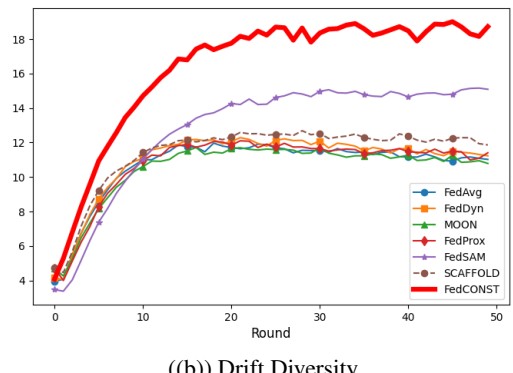

((b)) Drift Diversity

*Figure 3.* **Gradient variance and Drift diversity.** Our method reduces the gradient variance, thereby stabilizing the local training of client models. Simultaneously, it enhances drift diversity, ensuring that each client learns effectively even with constraints.

Table 2 shows how the application of constraints secures proper alignment, thereby sculpting a more convex and advantageous loss landscape for the global model. Furthermore, a negative trace value of the Hessian suggests convergence to a saddle point—a non-ideal scenario. Hence, Table 2 also demonstrates that the implementation of constraints contributes to a more convex loss landscape, steering the model away from saddle points towards optimal convergence.

Additionally, we introduce client consistency as another key metric, which quantifies the consistency of local models across clients. This consistency is defined as $\sum_{m \in M} \frac{|D_m|}{|D|} |\Delta w_m^k|_2^2$, where lower values indicate that client updates remain proportionally aligned, preventing excessive deviations in heterogeneous data environments.

all clients. This increased consistency contributes to a structured and predictable optimization trajectory, reinforcing the benefits of convexity-based alignment.

### 4.3.3. GENERALIZATION ON FEATURES OF GLOBAL MODEL

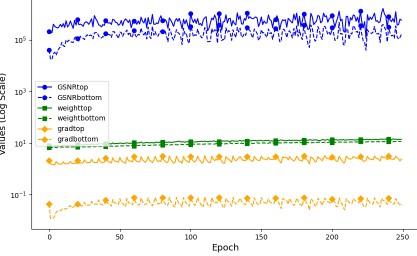

*Figure 5.* **Weight sizes and GSNR values.** Weight magnitudes and GSNR values sampled from the top 10% and bottom 10% of gradient update magnitudes across clients. Both metrics show correlation with gradient updates, suggesting their relevance to GSNR in the federated learning setting.

Our conjecture is that if already generalizable global feature is changed during local training of client, it harms generalization ability of FL process. To prove our hypothesis, we sampled top-10% and bottom 10% of gradient update size on changes. and we observe the GSNR value and weights size on corresponding feature to validate our hypothesis. And finally, we compared the GSNR value of vanilla FedAvg and our method indicating improved generalization performance.

First, Weight sizes corresponding to top 10% of Gradient update sizes have larger value than bottom 10%. Obser-

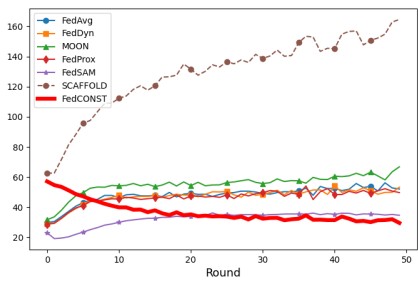

*Figure 4.* **Consistency.** Our method enhances the consistency among clients, by ensuring a more aligned learning experience across all clients.

The experimental results in Figure 4 confirm that our method significantly enhances client consistency compared to FedAvg, ensuring a more aligned learning experience across

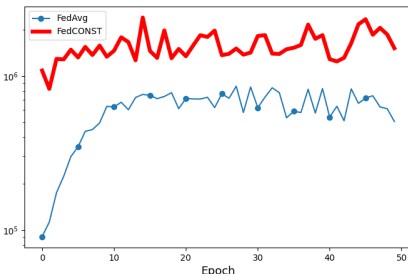

Figure 6. **The sum of GSNR values on a Client.** The sum of GSNR values on the initial local epoch of the proposed method is higher, indicating that features aligned with the global model are properly enhanced.

vation on Figure 5 indicates gradient update on client is associated with global generalizable feature, which harms generalization ability of global model. On GSNR values, top 10% also has larger values showing more direct impact of client training on generalization of global model.

After we impose our constraints that conserve common generalizable features, Figure 6 shows increase of GSNR values overall weight with constraints proving our method work as our intention.

## 5. Conclusion

In this research, we have introduced FedCONST, a novel FL algorithm that leverages convex constraints during optimization of client model for increasing generalization. This algorithm fosters stable local training on convex constraints, leading to a more generalizable global model through learning common features based on corresponding weight size of the global model. Our comprehensive experiments have shown that FedCONST not only stabilizes the learning process at the client level but also ensures consistent alignment to generalizable features. In various experimental settings, especially in the presence of highly heterogeneous data, FedCONST consistently outperformed existing algorithms.

## Acknowledgements

This work was supported by Korea Internet & Security Agency(KISA) grant funded by the Korea government(PIPC) (No.RS-2023-00231200, Development of personal video information privacy protection technology capable of AI learning in an autonomous driving environment)

## Impact Statement

Many studies on deep neural networks are conducted under the centralized learning paradigm with well-preprocessed datasets. However, real-world industrial data is often distributed, imbalanced, and noisy, making it challenging to apply academic research directly to practical settings despite the enormous potential of distributed data. Federated Learning (FL) is a promising technology to address this limitation, yet further improvements are needed for real-world deployment. This paper proposes a method to enhance FL performance by introducing a novel generalization perspective that mitigates its inherent limitations.

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

## A. Generalization Area and Implementation on Constraints

Figure 1 conceptually illustrates how aligning the aggregation step with the generalization area improves global generalization in FL. This motivates our use of convex constraints that stabilize both local training and aggregation. For our method to be effective, the constraint region must be contained within the generalization area. We argue that this condition is satisfied in practice, as the generalization area is sufficiently large under typical training regimes.

We provide an intuitive argument to suggest that the generalization area can be sufficiently large, based on the behavior of a $l$-th layer representation of a neural network:

$$\mathbf{e}_c^{(l)} = \phi\left((\mathbf{e}^{(l-1)})^\top \mathbf{W}_c^{(l)}\right), \tag{28}$$

where $\phi$ is an activation function (e.g., $\mathtt{tanh}$), and $\mathbf{e}_c^{(l)}$ denotes a representation within the generalization regime.

We consider whether the perturbed form

$$\phi\left(\mathbf{e}^{(l-1)^\top}(\mathbf{W}_c^{(l)} + \Delta\mathbf{W}_c^{(l)})\right) \approx \mathbf{e}_c^{(l)} \tag{29}$$

still holds under certain conditions.

**Case 1: Large $\|\mathbf{W}_c^{(l)}\|$ (Saturation Regime).** When $\|\mathbf{W}_c^{(l)}\|$ is large, the activation function saturates, and the output becomes relatively insensitive to $\Delta\mathbf{W}_c^{(l)}$. Thus, a wide range of perturbations can yield generalizable representations.

**Case 2: Small $\|\mathbf{W}_c^{(l)}\|$ (Linear Regime).** When $\mathbf{W}_c^{(l)}$ is small, the activation function behaves almost linearly:

$$\mathbf{e}_c^{(l)} \approx \mathbf{e}^{(l-1)^\top} \mathbf{W}_c^{(l)}. \tag{30}$$

We apply Chebyshev's inequality:

$$P\left(\left|\mathbf{e}_c^{(l)} - \mathbf{e}_{c,\text{goal}}^{(l)}\right| \geq \varepsilon\right) \leq \frac{\text{Var}(\mathbf{e}_c^{(l)})}{\varepsilon^2}. \tag{31}$$

Thus, when the variance is small, the representation stays close to the generalization target with high probability.

We note that gradient space alignment—especially orthogonal to $\mathbf{W}_c^{(l)}$—is helpful under our convexity assumptions, and that using OSGR-based preconditioning encourages high-GSNR updates that remain within the generalization zone.

Furthermore, generalization often means consistent loss across training and test—even if predictions are wrong—so the region itself is inherently wide.

## B. Implementation Details

### B.1. Training settings

**Hyperparameters.** In our experiments, we configured various algorithms with specific hyperparameters:

- MOON: $\mu = 0.01$, Temperature $= 1$

- FedProx: $\mu = 0.01$

- FedDyn: $\alpha = 1$

- FedSAM: $\rho = 1.0$

**Model Configuration.** We employed both the ResNet-18 and LenNet-5 architectures for our experiments. When applying our constraints, we removed the batch normalization layer to leverage the weight normalization (WN) effect. Additionally, biases were omitted from the models in our experiments, as they had only a minor effect on the overall model performance.

**Other Experimental Settings.** For the training parameters, we set the local learning momentum to 0.9, applied a weight decay of 1e-5, and used a batch size of 50. The learning rate was set to 0.01 for local training and 1.0 for global updates. All experimental evaluations were executed utilizing two Nvidia 3090 GPUs.

## B.2. Data Partitioning

**Datasets.** Our experiments were conducted using two well-known datasets: CIFAR-10 and CIFAR-100.

**Data Distribution Across Clients.** To simulate varying degrees of data heterogeneity across clients, we used Dirichlet distributions with different Alpha values: 0.5, 0.2, and 0.05. The distribution of data across clients, under these settings, is illustrated in Figure 9.

**Local Test Data.** For the evaluation of test loss on client data, we partitioned the data such that 10% of each client's data was reserved as local test data. This approach ensures that the test loss reflects the performance of the model under the specific data distribution of each client.

# C. Additional Experiments

## C.1. Ablation Study

**Impact of Constraints on Learning.** To understand the influence of each constraint on the learning process, we conducted an ablation study, examining accuracy graph and Hessian values.

**Accuracy Improvements.** As indicated in Table 3 (Accuracy), applying centralization and sphere constraints independently resulted in performance enhancements. The highest improvement was observed when both constraints were applied together.

| Orthogonal | Center | Performance (%) | $C_{convex}$ |
|:---:|:---:|:---:|:---:|
|  |  | 53.12 | 34.83 |
| o |  | 56.77 | 65.90 |
|  | o | 51.99 | 32.27 |
| o | o | 59.66 | 84.58 |

*Table 3.* Ablation study on the FedAvg algorithm assessing the impact of Sphere and Center optimization and their combined application on performance. The experiments were conducted using the LeNet-5 model on the CIFAR-10 dataset, with a Dirichlet distribution of 0.5. Performance is measured in terms of Top-1 accuracy (%) and $C_{convex}$ is defined as $|\lambda_{max}/\lambda_{min}|$.

**Learning Curves and Overfitting.** Observations from Figure 7 (Learning Curves) reveal that the application of sphere constraint helps prevent overfitting, contributing to more generalized local learning.

**Hessian Value Analysis.** Upon examining the Hessian values ( Table 3), we found that orthogonalization constraints tend to make the model more convex, implying better alignment among client models. centralization constraint, on the other hand, increases the speed of training of the model.

Our analysis indicates that both othogonalization and centralization significantly impact performance. Specifically, orthogonalization constraint align client models effectively, while centralization constraint boosts local learning, enhancing local learning capabilities.

## C.2. Hessian Values and Loss Landscape

**Analysis of Hessian Values.** Our observations, as detailed in Table 4, indicate an increase in the maximum eigenvalue despite the application of constraints. This challenges the conventional interpretation correlating the decrease in maximum eigenvalue with improved generalization, particularly in FL contexts. However, we noted a consistent increase in convexity with the application of constraints, suggesting that convexity might be a more reliable indicator in FL environments.

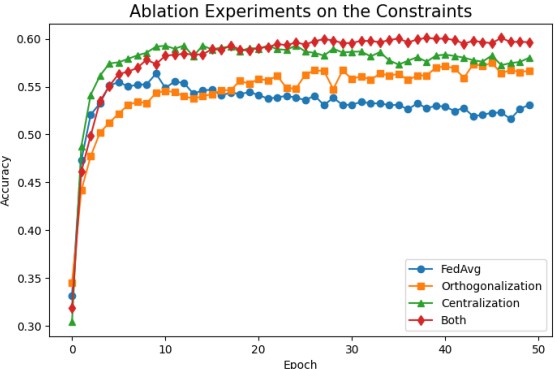

*Figure 7.* This figure represents the Top-1 accuracy per global epoch for the FedAvg algorithm under different constraint applications using the LeNet-5 model on CIFAR-10.

|           | w/o constraints | | constraints | |
|-----------|---------------------|------------|---------------------|------------|
| Algorithm | $\lambda_1/\lambda_5$ | $\lambda_1$ | $\lambda_1/\lambda_5$ | $\lambda_1$ |
| FedAvg    | 1.217 | 10.53 | 1.751 | 252.8 |
| FedProx   | 1.268 | 10.45 | 1.698 | 265.2 |
| MOON      | 1.252 | 9.51  | 1.718 | 258.5 |
| SCAFFOLD  | 1.278 | 10.05 | 1.594 | 177.6 |
| FedDyn    | 1.326 | 10.71 | 2.167 | 280.8 |

*Table 4.* In this analysis, we utilized the ratios $|\lambda_1/\lambda_5|$ and the maximum eigenvalue $\lambda_1$ of the Hessian matrix to assess the sharpness of the loss landscape. A lower value in these metrics typically indicates a flatter loss landscape, which is commonly associated with better generalization performance. Here, $\lambda_1$ represents the largest eigenvalue, while $\lambda_5$ denotes the fifth largest eigenvalue of the Hessian matrix. The table above demonstrates how the application of constraints leads to a sharper loss landscape, as indicated by these metrics.

**Convexity and Model Alignment.** The relationship between constraints and a more convex loss landscape is evident in Figure 10 (Loss Landscape). This convexity, indicative of effective model alignment, is further supported by Figure 11 (Eigen Spectral Density), which implies that constraints align the model towards more convex points.

These findings demonstrate the importance of considering convexity in the analysis of Hessian matrices in a FL setting. Unlike traditional settings where the focus is often on the maximum eigenvalue as a generalization indicator, our results highlight the significance of convexity in understanding model alignment and performance in FL. Therefore, observing convexity in the loss landscape and Hessian matrices could offer a more effective approach for analyzing and enhancing model performance in federated environments.

### C.3. More analysis on Learning Dynamics

**Cosine Similarity and Local Learning.** The Cosine Similarity ( Figure 15) analysis reveals that the application of constraints does not hinder the variability of cosine similarity. In fact, we observe an increased change on cosine similarity, suggesting that local learning is not restricted but appropriately regulated by the constraints. This indicates a balanced approach, where constraints guide the learning process without stifling the model's ability of local learning.

**GSNR and Model Generalization** The GSNR analysis highlights the impact of constraints on model alignment and generalization. Figure 14(a) presents the sum of GSNR values on a client at the initial local training step of each round. A higher GSNR at this stage suggests that FedCONST effectively leverages alignment with the global model to extract more generalizable features than FedAvg. Conversely, Figure 14(b) shows the sum of GSNR values at the final local training step of each round, where client models tend to drift from the global model. The observed decrease in GSNR values in FedCONST indicates that the constraints mitigate overfitting to client-specific data. This suggests that FedCONST maintains a more stable generalization process by preventing excessive reliance on localized information while preserving the overall adaptability of the model.

**Conjecture on Weight Magnitude and Generalization**    In Conjecture 1, we proposed that the magnitude of model parameters may reflect feature generality, and that preserving high magnitude weights could promote better generalization. To support this, we present a simple empirical analysis using t-SNE visualizations of global feature representations.

Specifically, we compare two variants of the global model: one where the bottom 20% of weights (by magnitude) are zeroed out, and another where the top 20% are removed.

We observe that excluding the bottom 20% of weights results in more clearly clustered and semantically aligned feature representations. In contrast, removing the top 20% of weights yields less structured outputs. This supports our conjecture that small-magnitude weights contribute more noise than signal, and that weight magnitude encodes meaningful signals about feature generality.

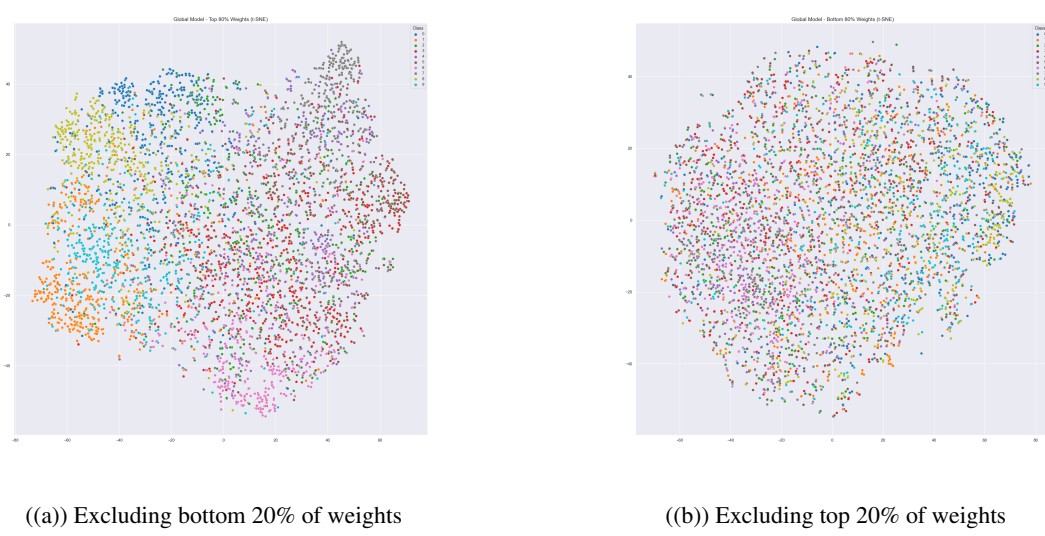

((a)) Excluding bottom 20% of weights                    ((b)) Excluding top 20% of weights

*Figure 8.* t-SNE visualization of feature representations from the global model. Removing small-magnitude weights (left) results in more clearly clustered and semantically aligned features, while removing large-magnitude weights (right) does not significantly improve semantic structure. This supports our conjecture that weight magnitude correlates with feature generality.

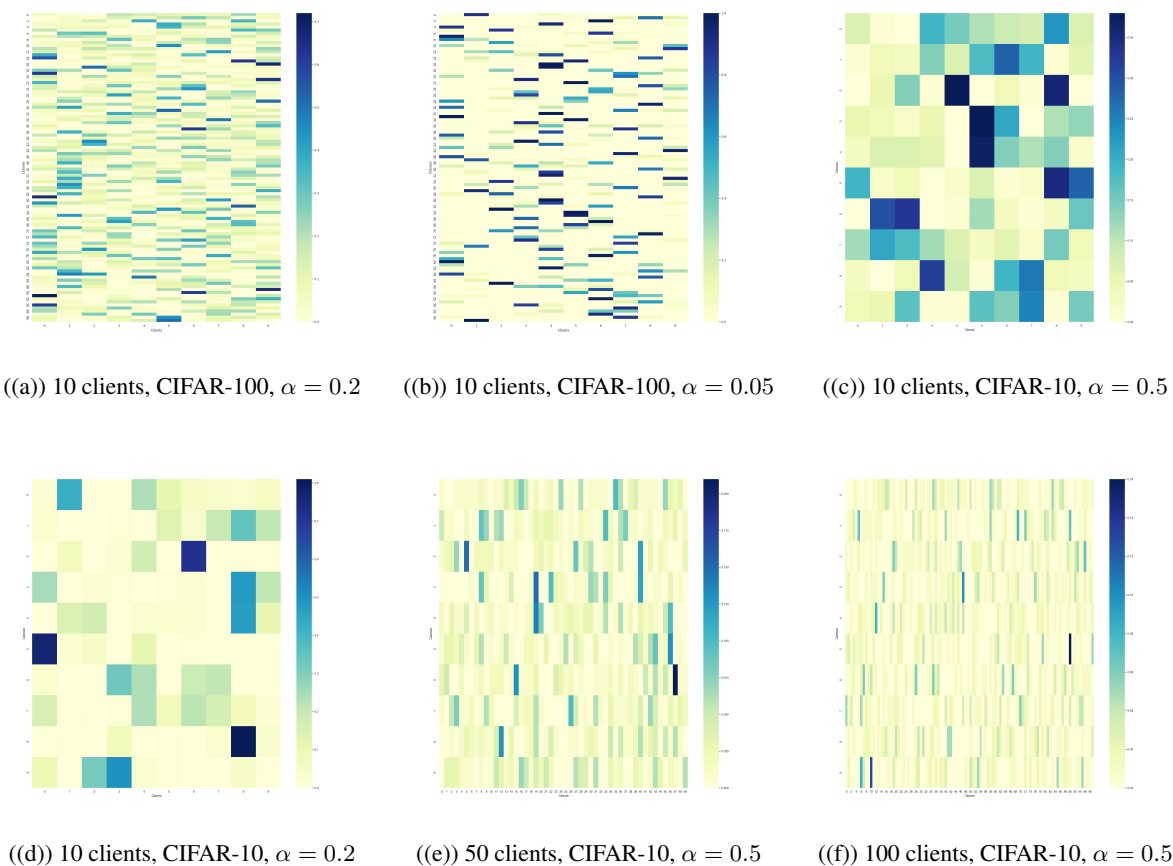

((a)) 10 clients, CIFAR-100, $\alpha = 0.2$    ((b)) 10 clients, CIFAR-100, $\alpha = 0.05$    ((c)) 10 clients, CIFAR-10, $\alpha = 0.5$

((d)) 10 clients, CIFAR-10, $\alpha = 0.2$    ((e)) 50 clients, CIFAR-10, $\alpha = 0.5$    ((f)) 100 clients, CIFAR-10, $\alpha = 0.5$

*Figure 9.* Example of data distribution according to (Client Number, Dataset, Dirichlet alpha). Each subfigure represents the data distribution under different client settings and Dirichlet $\alpha$ values.

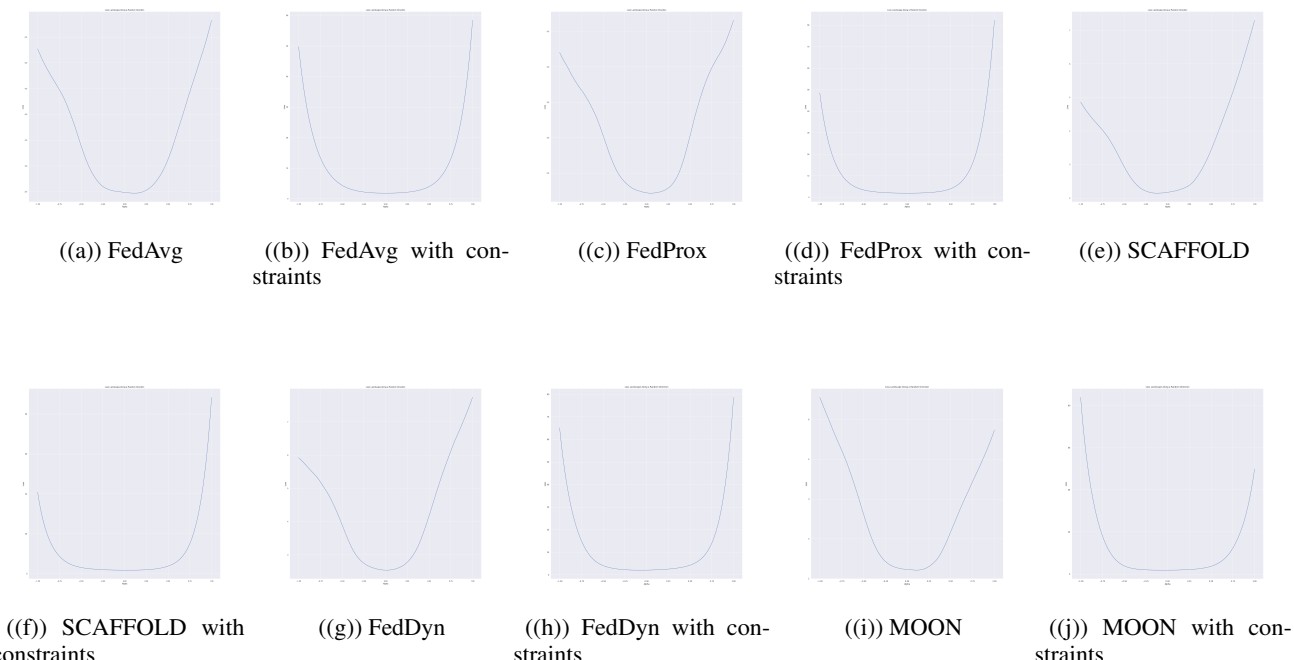

((a)) FedAvg ((b)) FedAvg with constraints ((c)) FedProx ((d)) FedProx with constraints ((e)) SCAFFOLD

((f)) SCAFFOLD with constraints ((g)) FedDyn ((h)) FedDyn with constraints ((i)) MOON ((j)) MOON with constraints

*Figure 10.* This figure presents the experimental results of the loss landscape for the ResNet-18 model in a cross-device setting of **??**. Noise was introduced to the weight $\|W\|$ in the form of a random vector $\epsilon$, scaled such that the ratio $\|\epsilon\|/\|W\|$ ranged from 0 to 1. The results demonstrate that applying constraints leads to a more convex loss landscape, indicating an enhanced generalization capability under these conditions.

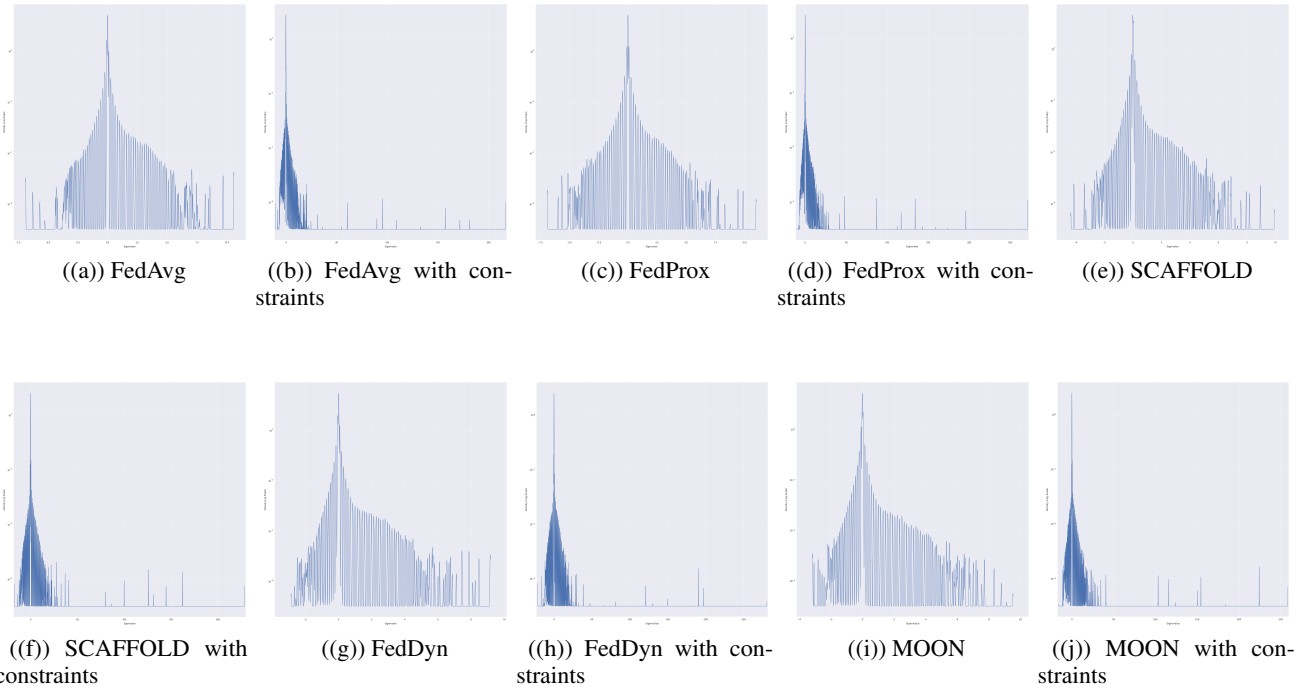

((a)) FedAvg ((b)) FedAvg with constraints ((c)) FedProx ((d)) FedProx with constraints ((e)) SCAFFOLD

((f)) SCAFFOLD with constraints ((g)) FedDyn ((h)) FedDyn with constraints ((i)) MOON ((j)) MOON with constraints

*Figure 11.* This figure illustrates the experimental results of the Eigen Spectral Density of the Hessian Matrix for the ResNet-18 model in a cross-device setting. Prior to the application of constraints, the density of negative eigenvalues is more significant, indicating the presence of saddle points in the loss landscape.

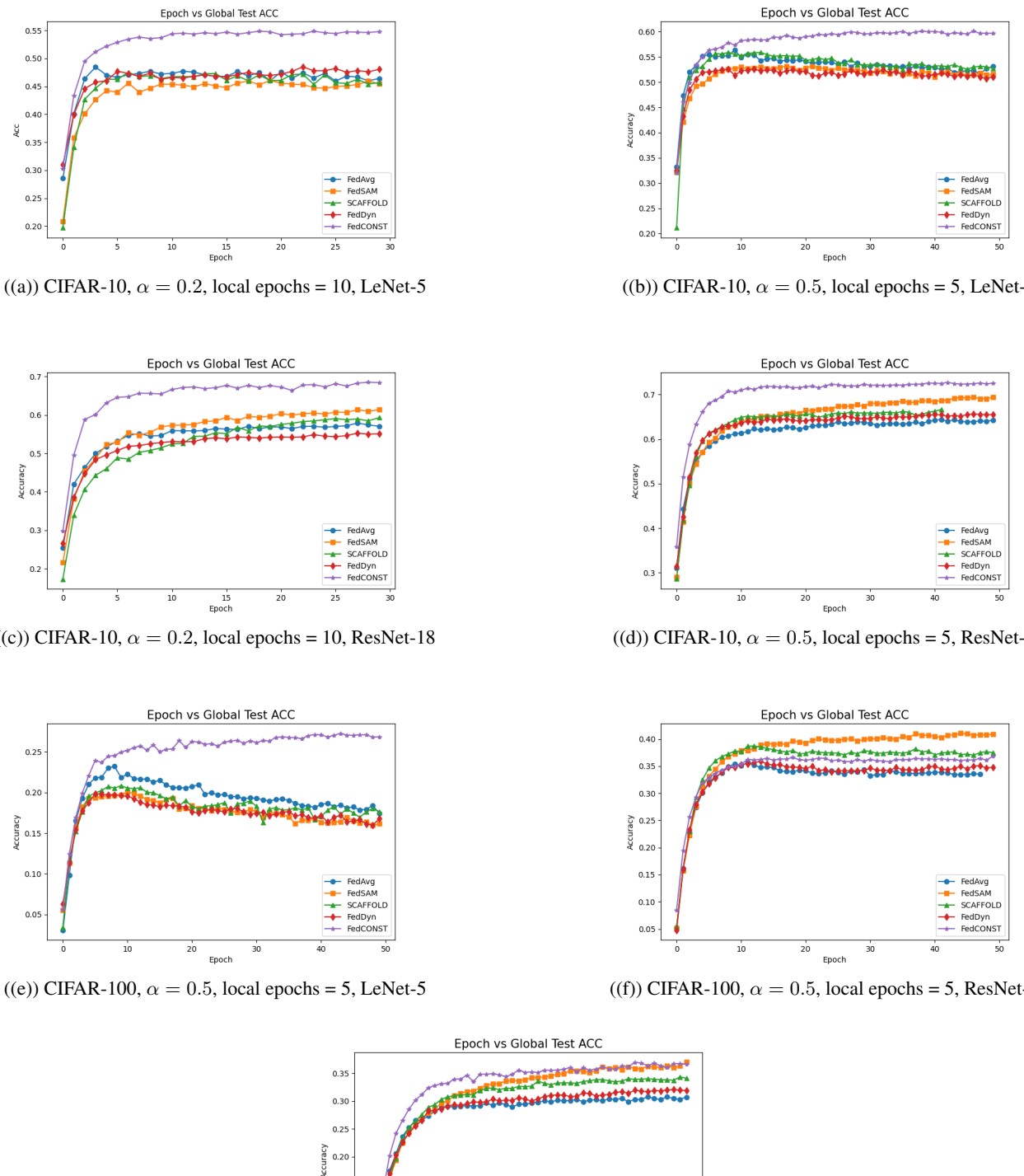

((a)) CIFAR-10, $\alpha = 0.2$, local epochs = 10, LeNet-5

((b)) CIFAR-10, $\alpha = 0.5$, local epochs = 5, LeNet-5

((c)) CIFAR-10, $\alpha = 0.2$, local epochs = 10, ResNet-18

((d)) CIFAR-10, $\alpha = 0.5$, local epochs = 5, ResNet-18

((e)) CIFAR-100, $\alpha = 0.5$, local epochs = 5, LeNet-5

((f)) CIFAR-100, $\alpha = 0.5$, local epochs = 5, ResNet-18

((g)) CIFAR-100, $\alpha = 0.05$, local epochs = 5, ResNet-18

*Figure 12.* Top-1 accuracy per global epoch for various algorithms conducted under cross-silo settings, with specific conditions (Dataset, Dirichlet alpha, local epoch, Model). This comparison highlights the performance variations across algorithms and the impact of different environments.

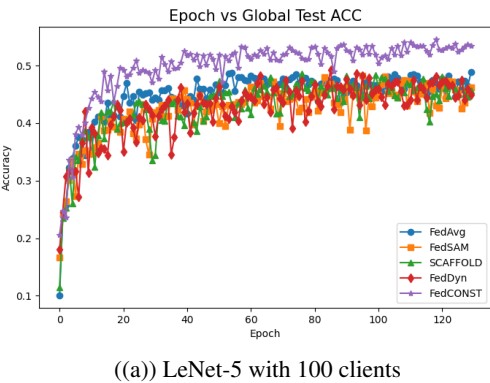

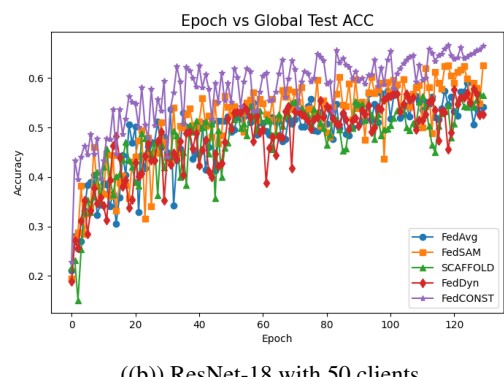

((a)) LeNet-5 with 100 clients

((b)) ResNet-18 with 50 clients

*Figure 13.* Top-1 accuracy per global epoch for various algorithms on CIFAR-10 under a cross-device setting, with a Dirichlet alpha of 0.5 and 10% client participation per round. Figure 13(a) shows results for LeNet-5 with 100 clients, and Figure 13(b) for ResNet-18 with 50 clients.

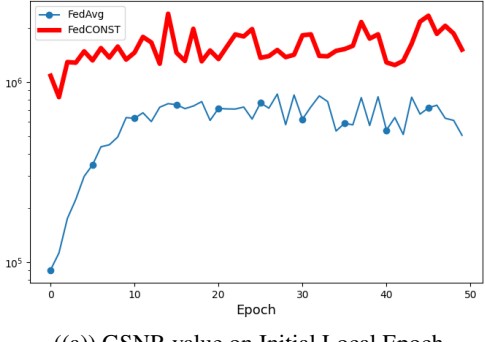

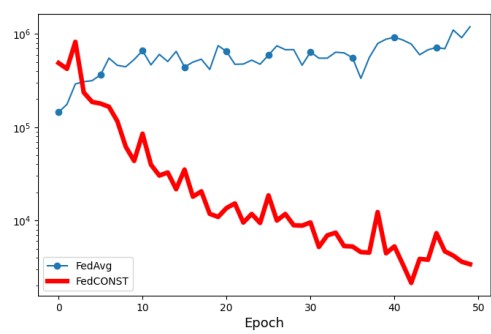

((a)) GSNR value on Initial Local Epoch

((b)) GSNR value on Final Local Epoch

*Figure 14.* Figure 14(a) shows the sum of GSNR values on a client at initial local training step of each round. When client model is aligned with global model, FedCONST harvests more generalizable features than FedAvg. Figure 14(b) shows the sum of GSNR values on a client at final local training step of each round. When client model is drifted from global model, there are less generalizable common features. Decrease in GSNR values on FedCONST indicates less overfitting to client data.

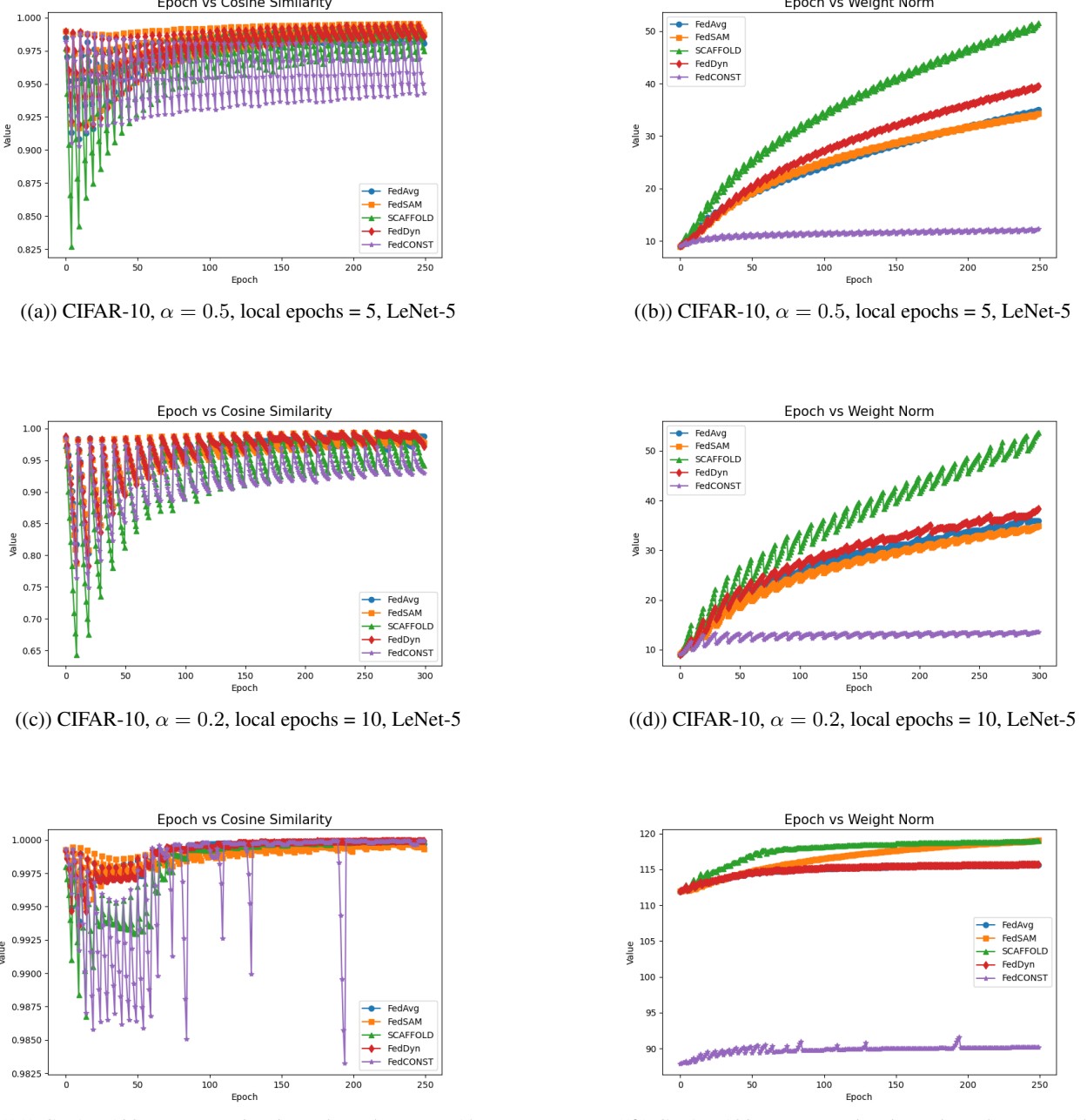

((a)) CIFAR-10, $\alpha = 0.5$, local epochs = 5, LeNet-5

((b)) CIFAR-10, $\alpha = 0.5$, local epochs = 5, LeNet-5

((c)) CIFAR-10, $\alpha = 0.2$, local epochs = 10, LeNet-5

((d)) CIFAR-10, $\alpha = 0.2$, local epochs = 10, LeNet-5

((e)) CIFAR-100, $\alpha = 0.05$, local epochs = 5, ResNet-18

((f)) CIFAR-100, $\alpha = 0.05$, local epochs = 5, ResNet-18

*Figure 15.* This figure displays the L2 norm of a client model and the cosine similarity between a client and the global model at each local epoch, for various algorithms implemented under cross-silo settings. The experiments were conducted with specific conditions (Dataset, Dirichlet alpha, local epoch, Model). The application of constraints consistently maintains the weight's L2 norm throughout training. In the case of ResNet-18, which includes batch normalization layers, the weight norm is naturally consistent. Notably, a large change in cosine similarity during training with constraints suggests that local learning is dynamically evolving and not overly restricted.

