# OpenReview forum: "Federated Learning for Feature Generalization with Convex Constraints"
_ICML.cc/2025/Conference — ICML 2025 poster_

### Official Review · Reviewer_ETbn · 2025-03-12

**Overall Recommendation:** 3

**Summary:**

For generalization in FL framework, this paper proposes FedCONST which impose more updates with larger probabilities to under-learned features by centralization and orthogonal constraints at clients. Various FL algorithms could get performance improvement with the proposed constraints.

## update after rebuttal
Thank you for your rebuttal. I have reviewed the authors’ rebuttals to all the reviews. I think most of the concerns are resolved in the rebuttal. Therefore, I want to increase my score to 3. Weak accept.

**Claims And Evidence:**

Positive correlation between weight magnitude and GSNR, gradient variance reduction, drift diversity are supported by experiments.

**Essential References Not Discussed:**

Most of the related papers are well cited.

**Experimental Designs Or Analyses:**

Experiments are somewhat small-scale. Federated domain generalization benchmark datasets such as PACS and OfficeHome would be better to show the effectiveness of the generalization.

**Methods And Evaluation Criteria:**

Evaluations of using various FL methods with the additional proposed convex constraints for cross-device and cross-silo conditions makes sense.

**Other Comments Or Suggestions:**

$w_i^l$ might be $w_c^l$ below Eq. (8)

**Other Strengths And Weaknesses:**

- Strengths
  - It is an interesting idea to include convex constraint to the local model updates by considering the aggregated global model.
  - Impressive performance improvement can be observed in Table 1.
- Weaknesses
  - Due to the centralization constraint, the aggregated weight exists within the convex constraint area. However, for the aggregated model to exist within the generation area, the convex constraint area should be within the generation area. I guess that the generalization area is generally smaller than the convex constraint area.
  - Some missing details and typos make it difficult to understand. E.g. formulations of center gradient and project gradient, ‘i' is used for sample or channel.
  - The experiments were conducted on small-scale datasets with limited model architectures (mainly conv networks). It would be better to evaluate the Conjecture 1 (Global weight magnitude is a reliably proxy for feature strength) with various network architectures.

**Questions For Authors:**

1) I am not sure why the convex constraints guarantee the aggregated global model belongs to the generalization area as shown in Figure 1. Could you explain it with more intuitive examples?
2) In Algorithm 1, what are the exact formulations of the Center gradient $C(g^k_{m,t})$ and Project gradient $P(g^k_{m,t})$. And why are center gradient and project gradient calculated serially?
3) Is the proposed method applicable to transformer architectures?

**Relation To Broader Scientific Literature:**

This paper proposes a federated domain generalization method by introducing convex constraints.

**Theoretical Claims:**

Theoretical analysis mainly based on “Understanding Why Neural Networks Generalize Well Through GSNR of Parameters, ICLR’20”

---

> ### Author Rebuttal · Authors · 2025-03-31
>
> ## Generalization Area
>
> We appreciate your insightful concern regarding the relationship between the convex constraint region and the generalization area. You raised an important point: if the generalization area is narrower than the constraint region, then the aggregated model could potentially drift away from the generalizable zone. In response, we offer an intuitive argument suggesting that the generalization area can, in fact, be sufficiently large.
>
> Consider the $l$-th layer representation of a neural network:
> $$
> \Phi^l_c = \sigma((\Phi^{l-1})^\top W^l_c)
> $$
>
> where $\sigma$ is the activation function (e.g., $\tanh$), and $\Phi^l_c$ denotes a representation within the generalization regime.
> We ask: under what conditions does
> $$
> \sigma((\Phi^{l-1})^\top (W^l_c + \Delta W^l_c)) \approx \Phi^l_c
> $$
> still hold?
>
> ### Case 1: Large $\|W^l_c\|$ (saturation regime)
>
> If the norm of $W^l_c + \Delta W^l_c$ is large, then the activation function saturates.
> In this case, $\sigma$ becomes relatively insensitive to $\Delta W^l_c$, so a wide range of perturbations still yield generalizable representations.
>
> ### Case 2: Small $\|W^l_c\|$ (linear regime)
>
> When weights are small, $\sigma$ behaves almost linearly:
> $$
> \Phi^l_c \approx (\Phi^{l-1})^\top W^l_c
> $$
>
> We can apply Chebyshev’s inequality:
> $$
> P(|\Phi^l_c - \Phi^l_{c,\text{gen}}| \geq \epsilon) \leq \frac{\text{Var}(\Phi^l_c)}{\epsilon^2}
> $$
>
> Here, the variance is small, so the output remains close to generalization with high probability.
>
> These cases show that the perturbation space $\Delta W^l_c$ that preserves generalization is often broad, especially when orthogonal to $W^l_c$—a behavior encouraged by our convex constraints.
>
> Furthermore, generalization often means consistent loss across training and test—even if predictions are wrong—so the region itself is inherently wide.
> Our formulation characterizes the generalization region by high GSNR parameters, which span a broad zone, especially early in training (Liu et al., ICLR 2020).
>
> Because our convex constraints encourage high-GSNR directions, they not only prevent drift but also promote convergence within the generalization zone.
>
> ---
>
> ## Implementation on Constraints
>
> We appreciate the reviewer’s observation. Indeed, when both constraints are affine (e.g., linear equalities), it is possible to compute a closed-form projection directly onto their intersection:
>
> $$
> \mathbf{x}^* = \mathbf{v} - A^\top (A A^\top)^{-1} (A \mathbf{v} - \mathbf{b})
> $$
>
> Where $A$ stacks the constraint vectors, and $\mathbf{b}$ contains target values.
>
> However, our implementation uses two-step sequential projection for simplicity and modularity:
>
> - Centering constraint:
>   $$
>   C(g_c^l) = g_c^l - \frac{1}{n_c^l} \mathbf{1}^\top g_c^l
>   $$
>
> - Projection constraint:
>   $$
>   P_{G^l_c}(g_c^l) = (I - G^l_c G^{l\top}_c) g_c^l
>   $$
>
> This modular approach keeps each step interpretable and separable.
> Nonetheless, we agree a single-step version might be more efficient and plan to explore this in the future.
>
> ---
>
> ## Transformer Architecture
>
> While we didn’t test on Transformers, our method is architecture-agnostic.
>
> Our constraints apply at the weight update level. In Transformers, this means we can project updates in attention matrices ($W^Q$, $W^K$, $W^V$) and feedforward layers ($W_1$, $W_2$), just as with CNNs.
>
> Because the projection is direction-aware and lightweight, it can be plugged into Transformer training pipelines without architectural changes.
> We plan to validate this in future work.
>
> ### Reference
> - Liu et al., "Understanding Why Neural Networks Generalize Well Through GSNR", ICLR 2020

---

### Official Review · Reviewer_9QZ9 · 2025-03-13

**Overall Recommendation:** 4

**Summary:**

This paper introduces FedCONST, a novel federated learning (FL) algorithm that addresses the challenges of generalization and overfitting in FL environments with heterogeneous data. By employing convex constraints based on the global model's parameter strengths, FedCONST adaptively modulates update magnitudes to prevent overemphasis on well-learned parameters while reinforcing underdeveloped ones. This approach not only stabilizes local training but also enhances feature transferability and robustness across diverse FL settings. The authors validate their method through extensive experiments on various datasets and model architectures, demonstrating state-of-the-art performance compared to existing FL techniques.

**Claims And Evidence:**

Yes. The author's claims regarding theoretical and experimental contributions are well-supported by concrete content.

**Essential References Not Discussed:**

No

**Experimental Designs Or Analyses:**

The experimental results effectively demonstrate the benefits of FedCONST in enhancing generalization and stability. However, in Figure 14(e), the large fluctuations in the L2 norm and cosine similarity for FedCONST toward the end of training raise concerns about potential instability.

**Methods And Evaluation Criteria:**

The proposed method, FedCONST , draws inspiration from Domain Generalization (DG) by leveraging Gradient Signal-to-Noise Ratio (GSNR) insights and applying convex constraints to enhance feature generalization in Federated Learning (FL). This approach is particularly effective in cross-silo settings, where it demonstrates significant improvements in generalization. However, the authors should provide a more detailed analysis of the computational overhead introduced by the method. Specifically:

- Theoretical Overhead : Discuss the additional computational cost of applying centralization and orthogonality constraints during local training.
- Practical Convergence Speed : Include convergence curves (e.g., test accuracy vs. global rounds) to assess whether the added complexity translates into faster or more robust convergence.

**Other Comments Or Suggestions:**

It is recommended to carefully review the figures and equations for potential typos. For example, in Figure 2 , the "Client - Side" section should likely be labeled as Client 1, Client 2, ..., Client N instead of the current notation, which appears to be inconsistent with standard representation.

**Other Strengths And Weaknesses:**

No

**Questions For Authors:**

No

**Relation To Broader Scientific Literature:**

FedCONST adapts the concept of GSNR from DG to FL, leveraging global weight magnitudes as a proxy for feature importance. This innovation aligns with recent efforts in model optimization research to enhance generalization under data heterogeneity, offering a computationally efficient and scalable solution tailored to FL's distributed nature. By stabilizing local training and preserving generalizable features during aggregation, FedCONST addresses key limitations of existing methods, such as overfitting and misalignment. These contributions inspire further exploration into constraint-based optimization strategies, particularly in scenarios involving sparse or imbalanced data distributions, and position the work as a meaningful advancement in both FL and general machine learning optimization paradigms.

**Theoretical Claims:**

While the paper introduces several formulaic expressions and provides a theoretical foundation for the proposed method, it lacks rigorous proofs for some of its core claims, particularly regarding convergence and generalization .

---

> ### Author Rebuttal · Authors · 2025-03-31
>
> We sincerely thank the reviewers for their constructive feedback and thoughtful suggestions.
>
> ## Theoretical Overhead:
> The proposed constraints—centralization and orthogonal projection—are implemented using simple linear operations with negligible computational overhead. Specifically, for a gradient vector of dimension $n$, each constraint introduces only $\mathcal{O}(n)$ additional computation per update during local training. No additional backpropagation steps or network modules are required. As a result, our method maintains computational efficiency and scalability within the federated learning setting.
>
> ## Practical Convergence Speed:
> As shown in Supplementary B.3, we provide convergence plots (test accuracy vs. global rounds) under various experimental settings. These results demonstrate that models trained with our constraint-based approach not only achieve better generalization but also converge more rapidly in terms of test accuracy. This indicates that the added constraints enhance both the learning dynamics and the final performance.

---

### Official Review · Reviewer_fYb2 · 2025-03-14

**Overall Recommendation:** 2

**Summary:**

This paper targets at addressing the generalization challenges in federated learning (FL). The authors propose FedCONST, an approach that adaptively adjusts update magnitudes based on the global model's parameter strength, preventing overemphasis on well-learned parameters and reinforcing underdeveloped ones. FedCONST employs linear convex constraints to maintain training stability and preserve locally learned generalization capabilities during aggregation. FedCONST aligns local and global objectives, mitigating overfitting and enhancing generalization across diverse FL environments, achieving state-of-the-art performance.

**Claims And Evidence:**

The claim on line 92 that "this paper provides theoretical and empirical analyses guarantee that the proposed method boosts generalization by imposing more updates with larger probabilities to under-learned features" is misleading, as the theoretical results in Theorem 2.1 appear to be directly adopted from existing research.

**Essential References Not Discussed:**

No.

**Experimental Designs Or Analyses:**

Yes, there are no outstanding issues in the designs and analyses of this paper.

**Methods And Evaluation Criteria:**

Yes.

**Other Comments Or Suggestions:**

Please refer to the weaknesses listed in the previous section.

**Other Strengths And Weaknesses:**

**Strengths:**

1. Extensive experiments on multiple benchmark datasets, including CIFAR-100, demonstrate the superior performance of the proposed method in diverse scenarios.

**Weaknesses:**

1. The essential motivation behind the proposed method is discussed in Conjecture 1, which, however, is not verified by either theoretical or empirical evidence in this paper. The soundness of this work would be improved if more evidence were provided to demonstrate the correctness of the claim in Conjecture 1.

2. The contributions of this work to the federated learning community are unclear. It appears that the theoretical results in Theorem 2.1 are directly adopted from an existing research paper. If this is the case, it would be more appropriate to title it as a Proposition rather than a Theorem.

3. The soundness of the evaluation section can be improved. For example, more datasets (e.g., EMNIST, ImageNet) should be considered to enhance the evaluation.

**Questions For Authors:**

Please refer to the weaknesses listed in the previous section.

**Relation To Broader Scientific Literature:**

The essential motivation behind the proposed method is discussed in Conjecture 1, which, however, is not verified by either theoretical or empirical evidence in this paper. As a result, the contributions of this work to the relevant community remain unclear.

**Theoretical Claims:**

Yes.

---

> ### Author Rebuttal · Authors · 2025-03-31
>
> We thank the reviewer for their thoughtful feedback. Your comments have helped us refine our intuition and better communicate the contributions of this work.
>
> ## Justification on Conjecture
>
> We appreciate your comment regarding the lack of direct evidence supporting *Conjecture 1*. In our paper, we proposed that the magnitude of global model parameters can serve as a proxy for feature generality, and that adjusting updates based on weight size could improve generalization. We provide further theoretical discussion in our response to Reviewer ETbn.
>
> To offer additional empirical support, we present a t-SNE visualization of feature representations from the global model (see [here](https://imgur.com/a/additional-validation-on-conjecture-UQB05mN); fully anonymized). In this analysis, we excluded the top 20% and bottom 20% of weights by magnitude to isolate the role of mid-range weights.
>
> Interestingly, removing the bottom 20% (i.e., the smallest-magnitude weights) resulted in *clearer clustering and more semantically aligned* representations. In contrast, removing the top 20% did not produce the same clarity.
>
> This observation supports our conjecture that small-magnitude weights may contribute more noise than signal, while mid-to-high magnitude weights are more aligned with generalizable features. This lends empirical credibility to our claim that **weight magnitude encodes meaningful signals about feature generality**, which justifies the use of convex constraints based on parameter norms to improve generalization.
>
> ## Contribution of this Work
>
> Thank you for the suggestion regarding terminology. We will revise terms like “Theorem 3.1” in the final version to avoid confusion. Our initial choice was to highlight the central theoretical result connecting constrained optimization with generalization.
>
> The main contribution of our work is to offer a new perspective on Federated Learning:
> The magnitude of global model parameters can implicitly encode information about the data distribution—information which can be harnessed even when local data is inaccessible.
>
> Building upon Theorem 2.1, we demonstrate that *simple convex constraints*, applied solely to global model weights, can influence generalization across clients. This is especially relevant in federated settings, where privacy constraints prevent access to raw data.
>
> Moreover, our work reframes the problem from *what* the model learns to *how* it learns—highlighting the role of directional constraints in implicit regularization and representation control in decentralized learning.
>
> Finally, we emphasize that our method is *lightweight and architecture-agnostic*:
> - No additional loss terms
> - No auxiliary models
> - No heavy computation
>
> This makes it highly practical for federated learning, where efficiency, privacy, and scalability are key concerns.

---

### Official Review · Reviewer_wAnr · 2025-03-14

**Overall Recommendation:** 2

**Summary:**

This paper proposes FedCONST, a federated learning (FL) framework, to boost generalization under heterogeneous client data distributions. Specifically, the authors adaptively modulate the magnitudes of updates based on the global model's parameter strength by applying convex constraints during client training. This prevents overemphasis on well-learned features while reinforcing underdeveloped ones. The authors further supported their approach with theoretical analyses and a series of experiments on different datasets, which validated its effectiveness in enhancing generalization.

**Claims And Evidence:**

Yes

**Essential References Not Discussed:**

No

**Experimental Designs Or Analyses:**

Yes

**Methods And Evaluation Criteria:**

Yes

**Other Comments Or Suggestions:**

No.

**Other Strengths And Weaknesses:**

Strengths:
1. The paper proposes an innovative and conceptually straightforward method to tackle overfitting and misalignment in FL.
2. It provides theoretical justification and extensive experimental validation for the strategy.
3. The writing is well-organized, with intuitive illustrations.

Weakness
1. The experimental evaluation relies solely on the CIFAR-10 and CIFAR-100 datasets. It is recommended that the experiments be supplemented with Tiny-ImageNet, as in some related works.
2. The baseline methods compared in the paper (e.g., FedAvg, FedProx) can validate the fundamental effectiveness of FedCONST. However, the selection of baseline models is somewhat outdated, failing to cover more recent federated learning methods, such as FedALA and FedFA.
3. Results in Table 1 report accuracy improvements but omit p-values or confidence intervals, leaving the significance of improvements unclear.
4. The proof assumes Gaussian-distributed weight updates (Equation 9), which may not hold in non-convex neural network optimization. Some discussion may be needed here.
5. The paper cites relatively few works and fails to cover many of the recent advances in the field. More recent relevant works need to be incorporated.

**Questions For Authors:**

In Section 3.2.2, while the projection matrix is constructed based on global model parameters, local model parameters are directly used in the derivation of Equation 13. Why does such a symbol substitution occur?

**Relation To Broader Scientific Literature:**

The paper situates itself well within the federated learning and domain generalization literature. It builds on established methods such as FedAvg, FedProx, and FedSAM while addressing known issues like client drift and feature misalignment. The theoretical inspiration drawn from GSNR-based dropout and domain generalization techniques is articulated, positioning the contribution as a natural yet innovative extension to the FL paradigm.

**Theoretical Claims:**

Yes

---

> ### Author Rebuttal · Authors · 2025-03-31
>
> We sincerely thank the reviewer for the constructive feedback and valuable suggestions. Below, we address each point in detail:
>
> ---
>
> ## Dataset diversity
> We appreciate your suggestion. While we agree that evaluating on larger datasets such as Tiny-ImageNet would further strengthen the experimental scope, we note that several recent works in the federated learning literature—including Wang et al. (CVPR 2024), Li et al. (ICML 2023), Qu et al. (ICML 2022), and Lee \& Yoon (ICML 2024)—also did not include Tiny-ImageNet in their evaluations.
>
> Instead, we extended our experiments to the EMNIST dataset. Preliminary results show the improvement using FedCONST, and we will include these results in the final version of the paper.
>
> ### EMNIST Accuracy Comparison
>
> | Method    | Accuracy (%) |
> |-----------|--------------|
> | FedAvg    | 83.54        |
> | FedCONST  | 85.10        |
> ---
>
> ## Comparison with recent FL methods
> We agree that comparing with more recent federated learning methods is important. While our main focus was on validating the fundamental effect of convex constraints through widely used baselines such as **FedAvg** and **FedProx**, we have additionally conducted experiments on recent algorithms, including **FedLAW**, **FedWon**, and **FedFA**, to strengthen the empirical comparison.
>
> The results below demonstrate that our proposed method performs competitively across both cross-device and cross-silo settings:
>
> | Algorithm       | Cross-Device CIFAR-10 (α=0.5) | Cross-Device CIFAR-10 (α=0.2) | Cross-Silo CIFAR-10 (α=0.5) | Cross-Silo CIFAR-100 (α=0.5) |
> |----------------|-------------------------------|-------------------------------|------------------------------|-------------------------------|
> | FedLAW (Wang et al., 2024) | 47.22                         | 45.15                         | 52.36                        | 17.80                         |
> | FedWon (Zhuang & Lyu, 2023)        | 35.80                         | 31.30                         | 39.31                        | 2.81                          |
> | FedFA (Qu et al., 2022)           | 48.14                         | 45.61                         | 50.95                        | 20.21                         |
> | **FedCONST (ours)**         | **54.28**                         | **54.79**                         | **59.66**                        | **26.86**                         |
>
> We will include these updated results in the final version.
>
> ---
>
> ## Statistical significance (p-values / confidence intervals)
> Thank you for pointing this out. We are currently computing **95% confidence intervals** based on repeated runs. Early results indicate that the performance improvements remain statistically significant. We will incorporate these confidence intervals in the final version of the results table.
>
> ---
>
> ## Gaussian assumption in the proof
> In our setting, local updates result from multiple gradient steps. By the Central Limit Theorem (CLT), their distribution tends to be Gaussian as the number of steps increases. Additionally, momentum-based optimizers accumulate gradients over time, reinforcing this tendency. We also assume the model has sufficiently many parameters per channel, allowing grouped parameters to be approximated as Gaussian. Thus, the Gaussian assumption is a practical and reasonable approximation for theoretical analysis in FL. We thank the reviewer for pointing this out and will include a more rigorous discussion on its validity in the revised version.
>
> ---
>
> ## Limited citations of recent work
> We appreciate the feedback and acknowledge that our related work section can be improved. We will revise it to incorporate more recent and relevant literature, particularly those that address generalization in FL and representation robustness.
>
> ---
> ## Clarification on Symbol Consistency in Equation 13
> In Equation 13, we used the same symbol for local updates and global model parameters since the local model starts from the global one. The weight update is computed as a delta from the global model, so the notational unification was intended for simplicity. However, we admit this may cause confusion, and we will revise the notation for clarity in the final version.
>
> ---
> ### References
> - Wang, Y., Fu, H., Kanagavelu, R., Wei, Q., Liu, Y., & Goh, R. S. M. (2024). *An aggregation-free federated learning for tackling data heterogeneity*. In CVPR.
> - Li, Z., Lin, T., Shang, X., & Wu, C. (2023). *Revisiting weighted aggregation in federated learning with neural networks*. In ICML.
> - Qu, Z., Li, X., Duan, R., Liu, Y., Tang, B., & Lu, Z. (2022). *Generalized federated learning via sharpness aware minimization*. In ICML.
> - Lee, T., & Yoon, S. W. (2024). *Rethinking the flat minima searching in federated learning*. In ICML.
> - Zhuang, W., & Lyu, L. (2023). *FedWon: Triumphing multi-domain federated learning without normalization*. arXiv:2306.05879.

---

### Decision · Program_Chairs · 2025-05-01

**Decision:**

Accept (poster)

**Comment:**

The paper proposes an approach using convex constraints to improve generalization in FL.
The approach depends on the conjecture that large weights indicate well-learned (strong) features while small weights signify weaker features requiring additional training.
The approach is orthogonal to many FL algorithms and can be applied to a variety of algorithms and the empirical results are significant.

While there was some concern about missing datasets or baseline methods, the paper seems to make a meaningful contribution that could apply more broadly and added onto prior approaches. Thus, the potential for impact is higher. In addition, the approach is simple and scalable.

The main suggestion is to increase the explanation and evidence for conjecture 1 and how this overall drives the paper. Also, more datasets and baselines would improve the evidence for the paper's contributions.